# *O*-Glycan-Dependent Interaction between MUC1 Glycopeptide and MY.1E12 Antibody by NMR, Molecular Dynamics and Docking Simulations

**DOI:** 10.3390/ijms23147855

**Published:** 2022-07-16

**Authors:** Ryoka Kokubu, Shiho Ohno, Hirohide Kuratani, Yuka Takahashi, Noriyoshi Manabe, Hiroki Shimizu, Yasunori Chiba, Kaori Denda-Nagai, Makoto Tsuiji, Tatsuro Irimura, Yoshiki Yamaguchi

**Affiliations:** 1Division of Structural Glycobiology, Institute of Molecular Biomembrane and Glycobiology, Tohoku Medical and Pharmaceutical University, Sendai 981-8558, Japan; 21711413@is.tohoku-mpu.ac.jp (R.K.); s.ohno@tohoku-mpu.ac.jp (S.O.); kuratani.hirohide@gmail.com (H.K.); y-takahashi@hosp.tohoku-mpu.ac.jp (Y.T.); manabe@tohoku-mpu.ac.jp (N.M.); 2Cellular and Molecular Biotechnology Research Institute, National Institute of Advanced Industrial Science and Technology (AIST), Ibaraki 305-8566, Japan; hiroki.shimizu@aist.go.jp (H.S.); y-chiba@aist.go.jp (Y.C.); 3Division of Glycobiologics, Intractable Disease Research Center, Graduate School of Medicine, Juntendo University, Tokyo 113-8421, Japan; k-denda@juntendo.ac.jp (K.D.-N.); t-irimura@juntendo.ac.jp (T.I.); 4Department of Microbiology, Hoshi University School of Pharmacy and Pharmaceutical Sciences, Tokyo 142-8501, Japan; m-tsuiji@hoshi.ac.jp

**Keywords:** antibody, docking simulation, glycopeptide, MD simulation, modeling, MUC1, NMR

## Abstract

Anti-mucin1 (MUC1) antibodies have been widely used for breast cancer diagnosis and treatment. This is based on the fact that MUC1 undergoes aberrant glycosylation upon cancer progression, and anti-MUC1 antibodies differentiate changes in glycan structure. MY.1E12 is a promising anti-MUC1 antibody with a distinct specificity toward MUC1 modified with an immature *O*-glycan (NeuAcα(2-3)Galβ(1-3)GalNAc) on a specific Thr. However, the structural basis for the interaction between MY.1E12 and MUC1 remains unclear. The aim of this study is to elucidate the mode of interaction between MY.1E12 and MUC1 *O*-glycopeptide by NMR, molecular dynamics (MD) and docking simulations. NMR titration using MUC1 *O*-glycopeptides suggests that the epitope is located within the *O*-linked glycan and near the *O*-glycosylation site. MD simulations of MUC1 glycopeptide showed that the *O*-glycosylation significantly limits the flexibility of the peptide backbone and side chain of the *O*-glycosylated Thr. Docking simulations using modeled MY.1E12 Fv and MUC1 *O*-glycopeptide, suggest that V_H_ mainly contributes to the recognition of the MUC1 peptide portion while V_L_ mainly binds to the *O*-glycan part. The V_H_/V_L_-shared recognition mode of this antibody may be used as a template for the rational design and development of anti-glycopeptide antibodies.

## 1. Introduction

It has been established that malignant transformation of cells involves abnormal glycosylation of the cell surface molecules [1]. Mucin1 (MUC1) was discovered as a carcinoma-associated mucin-like glycoprotein antigen and found to be peanut-agglutinin-reactive [2]. So far, many anti-MUC1 antibodies have been developed, and some of them are specific to cancer progression [3,4,5,6,7]. This specificity likely originates from the fact that abnormal *O*-glycosylation occurs at MUC1 on cancer cells. MUC1 is therefore considered the prime target of specific immunotherapy, including antibody-drug conjugates and CAR-T therapy [8,9]. Currently, some anti-MUC1 monoclonal antibodies are widely used as a clinical tool to detect and monitor breast cancer [10]. 

Information is currently rather limited on the precise epitope and binding specificity for many developed antibodies. So far, the binding modes of anti-MUC1 glycopeptide antibodies have mainly been revealed by X-ray crystallography. A crystal structure of SM3 in complex with MUC1 glycopeptide revealed that the antibody mainly recognizes the peptide part and the GalNAc residue points towards a solvent with a limited interaction with the SM3 antibody [11]. Another example is anti-MUC1 antibody AR20.5, and the co-crystal structure shows that the sugar moiety of the MUC1 glycopeptide does not directly contact the antibody [12]. It seems that *O*-glycosylation induces a preferred conformation, which is recognized for AR20.5 binding.

MY.1E12 (mouse IgG2a) is an antibody that was developed by immunizing mice with human milk fat membrane. It has been shown that this MY.1E12 antibody binds to MUC1 in an *O*-glycan-dependent manner [13,14]. Yoshimura et al. synthesized a series of MUC1 peptides with different glycosylation patterns as ligands for anti-MUC1 antibodies and investigated their affinity using an ELISA assay [15]. It was shown that MY.1E12 recognizes NeuAcα(2-3)Galβ(1-3)GalNAc only when attached to Thr8 of MUC1. This suggests that the binding of MY.1E12 is highly specific. Therefore, this antibody is a promising candidate for the development of cancer therapeutics. However, the structural basis of this unique binding mode has not yet been characterized.

In this study, we used NMR, MD simulations and docking simulations to analyze the mode of interaction between MUC1 *O*-glycopeptide and MY.1E12.

## 2. Results and Discussion

### 2.1. NMR Titration Study

First, we conducted NMR titration studies using MY.1E12 and MUC1 *O*-glycopeptides both to experimentally detect the interaction and to obtain information on the epitope region. We used three *O*-glycopeptides with different peptide chain lengths, i.e., MUC1 (27AA), MUC1(20AA) and MUC1 (9AA) (Figure 1). All three peptides have a single *O*-glycan (NeuAcα(2-3)Galβ(1-3)GalNAc) on Thr8. ^1^H-NMR signals originating from MUC1 *O*-glycopeptides were assigned by a series of 1D and 2D NMR experiments (Figure 2, Appendix A).

Upon the addition of MUC1 (27AA) to MY.1E12 solution, line broadening was observed in 1D ^1^H-NMR spectra for certain signals derived from MUC1 (27AA), e.g., NH signal of T8, GalNAc and NeuAc (Figure 2a, Appendix A). From this observation, the binding was experimentally confirmed in the solution between MY.1E12 and MUC1 (27AA). It was found that the NH signals from the C-terminus still showed a sharp signal in the presence of excess MY.1E12 (antibody:ligand = 1:0.5), suggesting that the C-terminal region of MUC1 (27AA) is not involved in the interaction with MY.1E.12. Line broadening of His side chains is also indicative of an antibody-binding region. There are two His residues in MUC1 (27AA), H5 and H25. H5 side-chain signals are broader than those of H25. This implies that the H5 side chain is at or near the antibody binding site, while that of H25 is not.

The binding was also monitored by 2D CLIP-COSY experiments to avoid signal overlapping (Figure 2b). In the presence of antibody, signals from the C-terminal region of the glycopeptide (S19, T20, A21 and V27) were clearly observed and sharp (Figure 2b). Therefore, it is likely that the C-terminal region of MUC1 (27AA) is not included in the binding epitope of this antibody. To quantitatively analyze the data, the peak heights of each signal in the CLIP-COSY spectra were measured and the ratio of peak height (MUC1+antibody/MUC1 alone) plotted (Figure 2c). This result supports the conclusion that the N-terminal region of the MUC1 glycopeptide is indeed involved in binding to antibodies.

We performed a similar NMR titration experiment using a shorter glycopeptide MUC1 (20AA) that still contained a putative epitope (Figure 3, Appendix A). We observed that the signal from V7 Hγ is significantly broadened, while the T20 Hγ signal remains sharp in the presence of an equimolar amount of antibody (antibody:MUC1 = 1:4) (Figure 3). This suggests that the C-terminal region of MUC1 (20AA) is less involved in MY.1E12 binding than the N-terminal region of the glycopeptide.

In addition, a 1D^−1^H NMR titration study of MUC1 (9AA) was performed, and binding was evident from the line broadening of NH signals from GalNAc and Thr8 and the acetyl signals from NeuAc, GalNAc and the N-terminus (Appendix A). Overall, we established that antibody binding occurs near the *O*-glycosylation site in three different MUC1 peptides (Appendix A). Taken together, NMR titration experiments suggest that MY.1E12 simultaneously recognizes the *O*-glycan and the peptide region near the *O*-glycosylation site of the MUC1 glycopeptide.

### 2.2. MD Simulations of MUC1

The conformational dynamics and role of the *O*-glycan exhibited by the MUC1 polypeptides were studied by performing MD simulations of *O*-glycosylated MUC1(9AA), unglycosylated MUC1 (9AA), and *O*-glycan alone (NeuAcα(2-3)Galβ(1-3)GalNAc). 

First, a comparison was made between the MD simulations from *O*-glycosylated and unglycosylated MUC1(9AA). For the peptide backbone, φ and ψ torsion angles were defined according to the IUPAC definition, and χ_1_ of Thr was defined as N-Cα-Cβ-O. It was found that the torsion angle distributions of ψ and χ_1_ are limited in the presence of *O*-glycan at Thr8 (Figure 4 and Appendix A). This observation is consistent with previous reports, showing that *O*-GalNAc modification restricts the flexibility of polypeptides through intramolecular GalNAc-peptide interactions [16,17,18].

Next, we examined the conformation of the *O*-glycan and the effect of peptide conjugation. The torsion angles of glycosidic linkages are as follows: NeuAc-Gal linkage is defined by φ (O-C2-O-C3′) and ψ (C2-O-C3′-C2′) and Gal-GalNAc linkage is defined by φ (O-C1-O-C3′) and ψ (C1-O-C3′-C2′). From our MD simulation, the torsion angle of the NeuAc-Gal linkage was φ = 49° ± 14 and ψ = −127° ± 41. For the Gal-GalNAc linkage, the angles were φ = −44° ± 31 and ψ = −148° ± 43 (Figure 5). The distribution of φ and ψ torsion angles between NeuAc and Gal residues were previously reported as φ = 69° ± 14 and ψ = −125° ± 16 [20]. The distribution of torsion angles between Gal and GalNAc residues are reported in the glycan fragment database as φ = −77° ± 18 and ψ = −150° ± 49 [21]. Therefore, the distributions of torsion angles from the MD simulation are rather consistent with previous reports. By comparing the glycosidic torsion angle distributions of *O*-glycosylated MUC1(9AA) and *O*-glycan alone, it can be concluded that the conformational dynamics of the *O*-glycan is slightly restricted in the presence of MUC1 polypeptide. This is consistent with the MD simulation showing that *O*-glycosylation limits the conformational dynamics of the MUC1 polypeptide (Figure 4, Appendix A).

### 2.3. Modeling of MY.1E12

To perform the docking study, we built a 3D model of the MY.1E12 Fv domain using Discovery Studio 2021. CDRs were identified using the Annotate Sequence tool in the Discovery Studio 2021 (Figure 6a), and the numbering scheme was based on IMGT [22]. A chimeric antibody (PDB ID: 3MBX IgG_1_) [23] was used as the template. It comprises an H chain, which is an anti-human IL-13 antibody (IgG), and an L chain, which is an anti-human EMMPRIN antibody (IgG). MY.1E12 shares a CDR sequence identity with these templates of 66.7%. A Ramachandran plot of the 3D model (Figure 6b,c) [24] shows that most of the amino acid residues except for Gly are located within the favored region, validating the 3D model in terms of the main chain dihedral angles.

### 2.4. Docking of MUC1 Glycopeptide to MY.1E12

Docking poses of MUC1 *O*-glycopeptide and MY.1E12 antibody as well as that of MUC1-MY.1E12 were built using ZDOCK software [25]. In the latter case, MUC1 (9AA) is known to bind to MY.1E12. Docking was performed under conditions such that the entire ligand and CDRs of MY.1E12 are involved in binding.

Since ZDOCK applies a rigid docking procedure, pseudo-flexible docking was performed using three MUC1 conformers that were extracted from the MD trajectory. The conformers were chosen at the simulation times of 8 ns, 9 ns, and 10 ns, and 30 docking poses were obtained. Of these, 10 (33.3%) were categorized into one group sharing a similar docking topology (Figure 7a). Solvent accessibility (ASA) was calculated from these 10 docking poses to identify the binding sites of the receptor and ligand. For the analysis of binding sites (epitope and paratope), solvent accessibility was calculated in the presence and absence of the binding partner. The results show that MY.1E12 uses CDR H1, H2 and H3 for binding to MUC1 *O*-glycan, while CDR L1, L2, L3 and H3 are involved in binding to the MUC1 peptide (Figure 7b). Paratope analysis shows that the N-terminal region of the MUC1 peptide (A1-S6) interacts with the V_H_ domain, while NeuAc is recognized by the V_L_ domain (Figure 7c). This NeuAc-V_L_ interaction is consistent with previous reports that sialic acid is essential for binding to MY.1E12 [14,15]. A 2D plot analysis indicates that heavy-chain CDR loops (H1, H2 and H3) interact with the peptide region, while the light chain CDR1 loop (L1) binds to the glycan part (Figure 7d).

The electrostatic potential of MY.1E12 Fv was calculated for one of the docking poses. The MUC1 glycan contains a negatively charged terminal NeuAc residue which may interact with a positively charged residue(s) on the CDR of MY.1E12. Indeed, the L1 loop exhibits a positively charged area associated with a lysyl residue located near the NeuAc residue (Figure 8).

## 3. Materials and Methods

### 3.1. NMR Analysis

Three MUC1 *O*-glycopeptides—MUC1 (27AA), MUC1 (20AA), and MUC1 (9AA)—were prepared with the sequences, APPAHGVT^8^SAPDTRPAPGST-OH, APPAHGVT^8^SAPDTRPAPGST-OH, and Ac-AHGVT^8^SAPD. Glycoprotein-*N*-acetylgalactosamine 3-β-galactosyltransferase 1 (dC1GalT) and CMP-*N*-acetylneuraminate-β-galactosamide-α-2,3-sialyltransferase 1 (ST3Gal1) were used to build the NeuAcα(2-3)Galβ(1-3)GalNAc glycan onto Thr8 [15]. NMR analyses were performed using JNM-ECZ600R/S1 600 MHz spectrometer equipped with a ROYAL probe (JEOL, Tokyo, Japan), or AVANCE III 600 MHz spectrometer equipped with a TXI probe (Bruker, Billerica, MA, USA). The probe temperature was set to 278 K. MUC1(9AA) (1.6 mg) and MUC1(20AA) (1.7 mg) were dissolved in 600 µL of 20 mM sodium phosphate buffer, pH 6.8 (H_2_O:D_2_O = 9:1). MUC1(27AA) (0.48 mg) was dissolved in 300 µL of 25 mM sodium phosphate buffer, pH 6.8 (H_2_O:D_2_O = 9:1). ^1^H chemical shifts were reported by reference to the internal standard of 4,4-dimethyl-4-silapentane-1-sulfonic acid (DSS, 0 ppm). NMR chemical shifts of MUC1 were assigned by analyzing 1D-^1^H, 2D DQF-COSY, CLIP-COSY, HOHAHA, and NOESY spectra. ^1^H titration study was performed with 1024 scans for MUC1(27AA) and 128 scans for MUC1(20AA and 9AA). Suppression of water signal was performed using a watergate sequence. The ^1^H pulse length was typically 8–10 µs. NMR data processing was performed using Delta5. 3. 1 (JEOL, Tokyo, Japan), and NMR spectral analyses were performed using Mnova 14. 1. 1 (Mestrelab Research, Santiago, Spain). For MUC1 (9AA) and MUC1 (20AA) experiments, the concentration (binding site) of MY.1E12 (mouse IgG2a, *M*_W_ = 150,000) was 4.3 µM, dissolved in 600 µL of 20 mM sodium phosphate buffer, pH 6.8 (H_2_O:D_2_O = 9:1). For MUC1 (27AA), the concentration (binding site) of MY.1E12 was 25 µM, dissolved in 500 µL of 25 mM sodium phosphate buffer, pH 6.8 (H_2_O:D_2_O = 9:1).

### 3.2. MD Simulation of MUC1

Coordinates of MUC1 glycan were created using Carbohydrate Builder in GLYCAM (https://dev.glycam.org/) (accessed on 6 June 2021). Peptide coordinates were created using Discovery Studio 2021 [24] The peptide and glycan were then attached using a Glycoprotein Builder in GLYCAM. The coordinates of MY.1E12 were created by homology modeling. CHARMm [26] was assigned as the force field. Simulation time was set to 10 ns. Explicit periodic boundary was used as the solvation model. Orthorhombic cell shape was used in explicit periodic boundary solvation model. The minimum distance from periodic boundary was set to 7.0 Å. For each MUC1 coordinate, 439–1045 water molecules were explicitly placed and TIP3 [27] was used as the force field template. Minimization of the initial coordinate was done in two steps. The first step eliminated the distortion of the entire structure with the steepest descent algorithm. In the second step, minimization was performed with adopted basis Newton–Raphson (NR). Heating was carried out at 310 K. After equilibration, the time step was set to 2 fs and NAMD was carried out under an *nPT* ensemble [28].

### 3.3. Modeling of MY.1E12 Fv Domain

The 3D structure of the antibody was generated by a homology modeling technique. The amino acid sequences were as follows, with CDR underlined:

MY.1E12 V_H_

QVTLKESGPGILQPSQTLSLTCSFSGFSLSTLGMGVSWIRQPSGKGLEW-LAHIYWNDDKHYNPSLKSRLTISKDSSINQVFLRITTVDTADAATYYCART NYYGSSYDYWGQGTTLTVSS

MY.1E12 V_L_

DIVMTQSPSSLTVTAGEKVTMSCKSSQSLLHSGNQKNYLTWYQQKPGQPPKLLIYWTSTRESGVPDRFTGSGSGTDFTLTISSVQAEDLAVYYCQNDYSY PFTFGSGTKLEIKR

Identification of suitable homologous template structures for the modeling of the target protein was carried out using the tool BLAST. CDRs were identified using the Annotate Sequence tool. Then, Identify Framework Templates was used to search for candidate templates. CDR numbering scheme was based on IMGT [22]. As a result, chimeric antibodies against IL-13 and EMMPRIN (PDB ID: 3MBX) were adopted as templates. Modeling was performed with the Model Antibody Framework tool. The Model Antibody Loop was subsequently performed to rebuild the CDR.

### 3.4. Docking Simulation of Glycopeptide and Antibody

Docking simulation of antibody–glycopeptide complex was performed using ZDOCK. For MUC1-MY.1E12 docking, MY.1E12 was used as the receptor and MUC1 (9AA) as the ligand. Stable structures for docking were derived from those at the end of MD simulation. ZDOCK is a rigid body docking algorithm, and to create alternative ligand structures we selected three MUC1 conformers in the middle of the MD simulation. The conformers used were those at 8 ns, 9 ns, and 10 ns. In the docking simulations, all glycan regions and all peptide regions were considered as active sites based on the NMR and MD simulation results. The active site of MY.1E12 was defined as the CDR region, and the other sites were defined as blocking sites. The contact surface area was calculated using ASA.

## 4. Conclusions

To gain an understanding of the structural basis of the *O*-glycan-dependent interaction between MUC1 and MY.1E12 antibody, we used several approaches. NMR titration locates the epitope to the *O*-glycan and nearby amino acid residues. MD simulation suggests *O*-glycosylation limits the conformational flexibility of the *O*-glycosylation site. In silico docking implicates both the *O*-glycan and peptide of the MUC1 ligand in binding to the MY.1E12 antibody. The elucidation of the likely mode of recognition will help develop novel glycopeptide-specific antibodies with desired sequence specificity. We are continuing to clarify the detailed binding mode of this antibody.

## Figures and Tables

**Figure 1 ijms-23-07855-f001:**
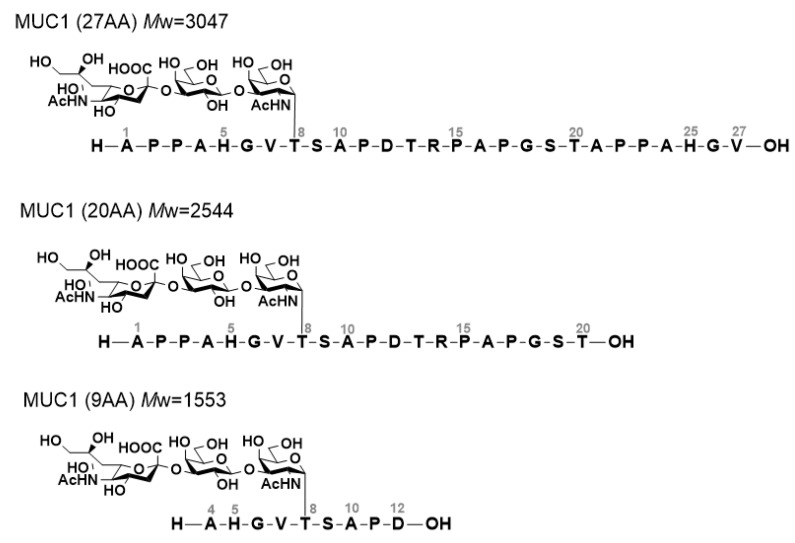
MUC1 *O*-glycopeptides (27AA, 20AA and 9AA) used in the NMR titration study.

**Figure 2 ijms-23-07855-f002:**
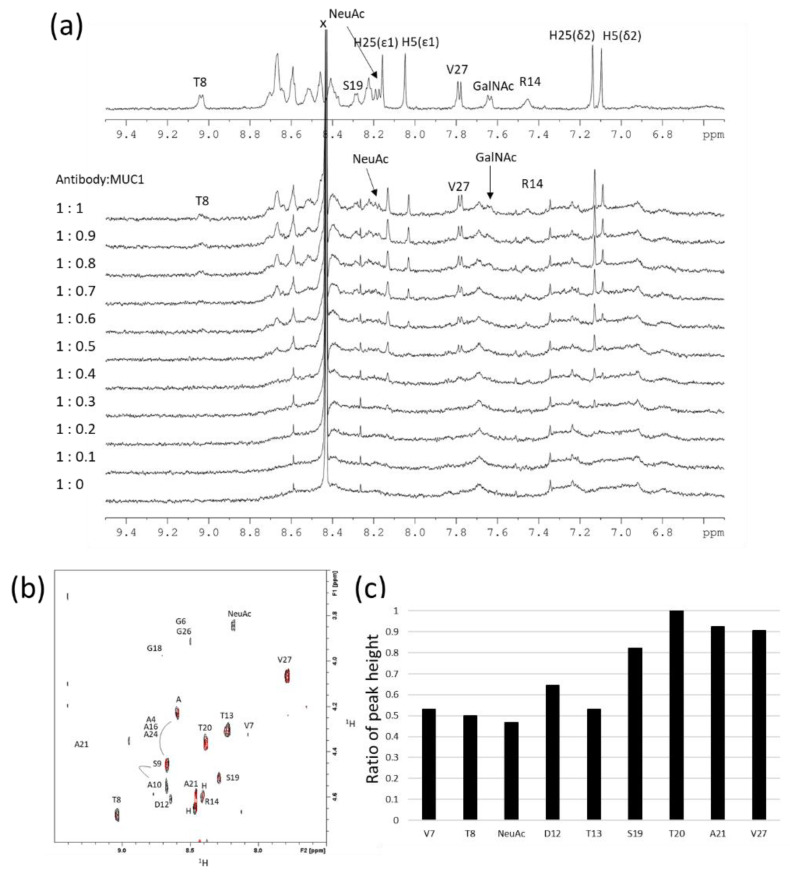
NMR titration experiments using MUC1 (27AA) and MY.1E12. (**a**) 1D-^1^H NMR spectra (amide NH region) of MY.1E12 with increasing amount of MUC1 (27AA). Molar ratios (binding site) are indicated for each spectrum. x: impurities. (**b**) 2D CLIP-COSY spectra (NH-H*α* region) of 25 µM MUC1 (27AA) in the absence (black) or presence (red) of equimolar amount of MY.1E12. A4, A16, A24, S9 and A10 signals were not uniquely assigned. (**c**) Ratio of peak height (MUC1+antibody/MUC1 alone) for each NH-H*α* cross peak in CLIP-COSY spectra. The ratios are normalized against the highest value (T20). All experiments were performed on a 600 MHz spectrometer at 278 K.

**Figure 3 ijms-23-07855-f003:**
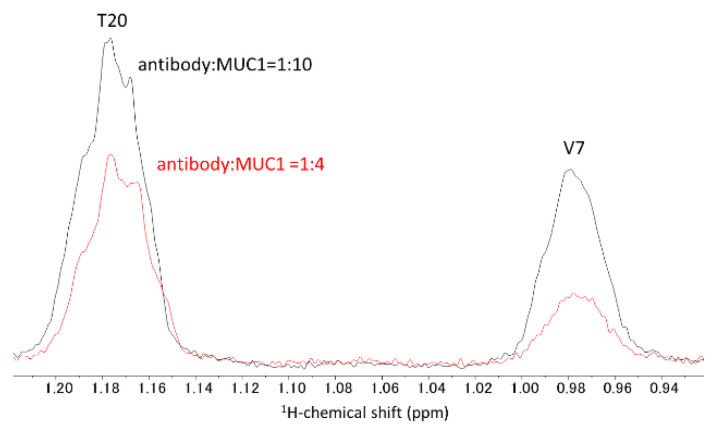
NMR titration experiments using MUC1 (20AA) and MY.1E12. The methyl region is selected to show T20 and V7 methyl signals (black; antibody:MUC1 = 1:10, red; antibody:MUC1 = 1:4). The experiments were performed on a 600 MHz spectrometer at 278 K.

**Figure 4 ijms-23-07855-f004:**
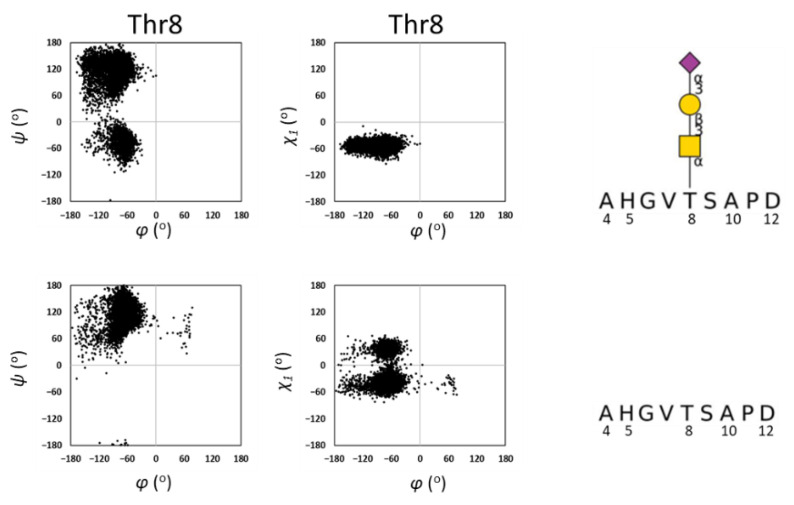
Distribution of dihedral angles of Thr8 of MUC1 obtained by MD simulation. *O*-glycosylated MUC1 peptide (**top**) and unglycosylated peptide (**below**) were simulated. Symbol nomenclature for glycans (SNFG) is used for presenting *O*-glycan [19].

**Figure 5 ijms-23-07855-f005:**
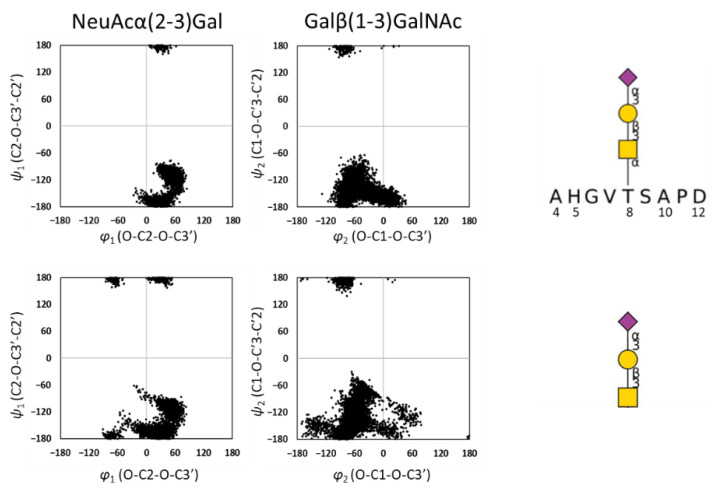
Distribution of dihedral angles of *O*-glycan in MUC1 obtained by MD simulation. *O*-glycan presented on MUC1 peptide (**top**) and *O*-glycan alone (**below**) were simulated.

**Figure 6 ijms-23-07855-f006:**
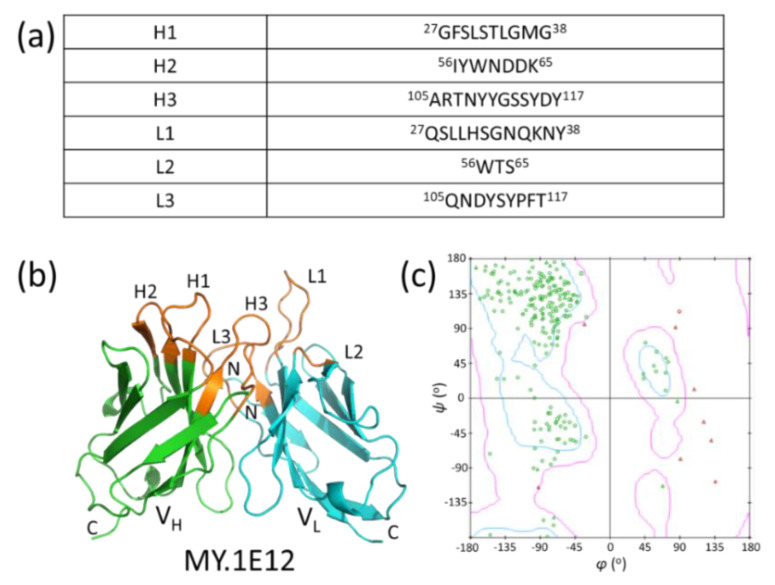
Construction of a 3D model of MY.1E12 Fv region. (**a**) Amino acid sequences of MY.1E12 CDRs defined by IMGT. (**b**) A 3D model of MY.1E12 Fv fragment generated by homology modeling. V_H_: variable region of H chain, V_L_: variable region of L chain. Orange: CDR. (**c**) Ramachandran plot of the modeled MY.1E12 Fv region. Gly is shown as triangles, Pro squares, and the other amino acid residues as circles. Green symbols are within the energetically favorable region (boundary line in pink) and red symbols are located outside the region.

**Figure 7 ijms-23-07855-f007:**
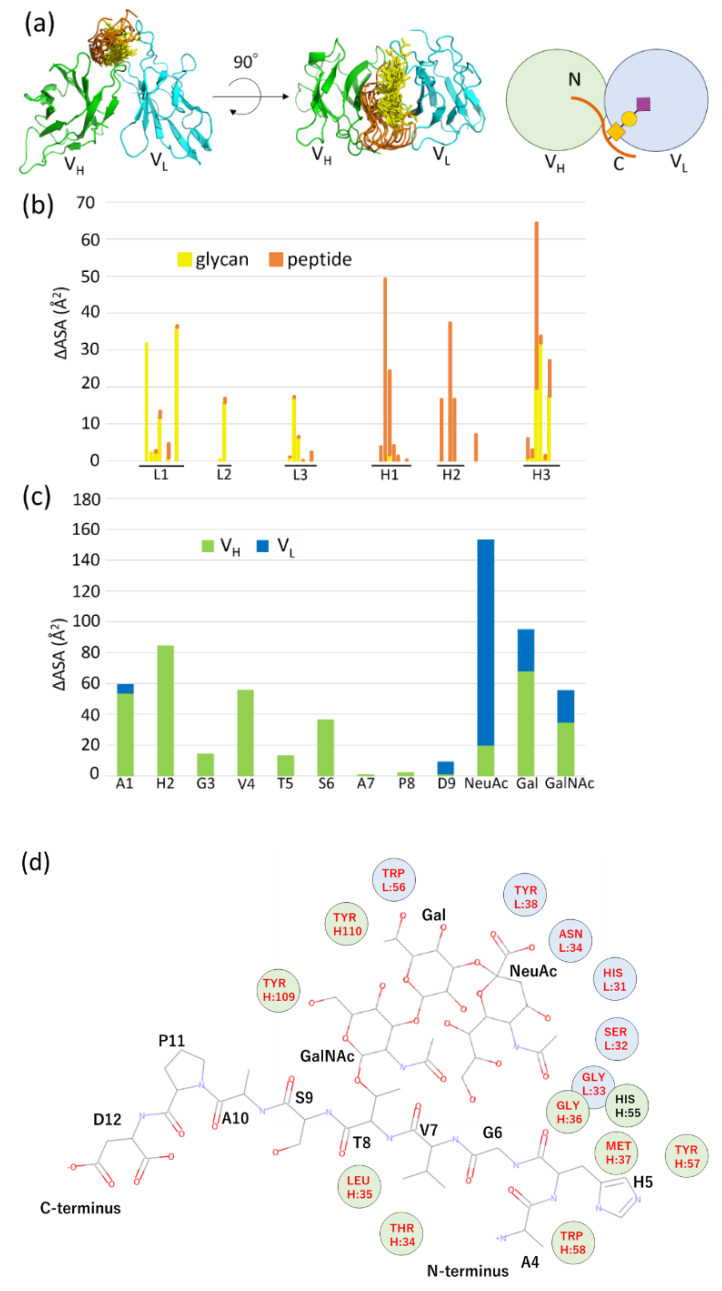
Docking simulations and analysis of MUC1-MY.1E12 binding site. (**a**) Ten selected docked poses that share a similar binding mode. V_H_ is shown in green, V_L_ in light blue, MUC1 *O*-glycan in yellow, and the peptide region in orange. Schematic drawing of the docking pose is shown on the right. (**b**) MUC1-binding region (epitope) of MY.1E12 based on difference in solvent accessibility (ΔASA) in the presence and absence of MUC1 ligand. Contributions of MUC1 glycan and the peptide portion to antibody binding are shown separately in yellow and orange, respectively. (**c**) Antibody-binding region (paratope) of MUC1 based on difference in solvent accessibility (ΔASA) in the presence and absence of antibody. Contributions of V_H_ and V_L_ to MUC1 binding are shown separately in green and blue, respectively. (**d**) Schematic 2D plot showing the interaction between MUC1 glycopeptide and antibody MY1E12. V_H_ is shown in green, V_L_ in light blue, CDR in red.

**Figure 8 ijms-23-07855-f008:**
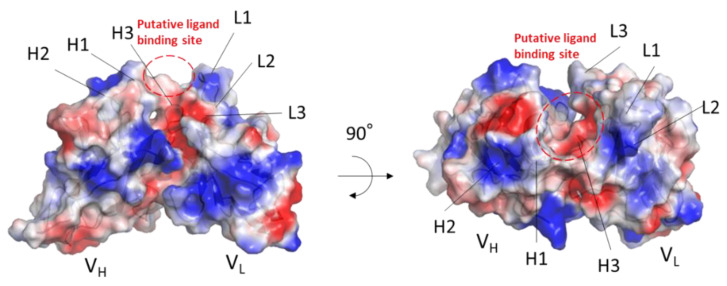
Electrostatic surface potential of MY.1E12 Fv domain. The surface model of MY.1E12 Fv is colored according to the electrostatic surface potential (blue, positive; red, negative; scale from −50 to +50 kT/e). Putative ligand-binding areas are circled with red dotted line. Figures were prepared using PyMol software.

## Data Availability

Not applicable.

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
