# Peer review of "O-Glycan-Dependent Interaction between MUC1 Glycopeptide and MY.1E12 Antibody by NMR, Molecular Dynamics and Docking Simulations"

_ijms, 2022, doi:10.3390/ijms23147855_

Round 1

Reviewer 1 Report

Though O-glycan-dependent binding was observed in previous studies, I would recommend comparing the binding of MUC1 with and without O-glycan by NMR.

If compared the spectra of the bound state of MUC1 in 3 different lengths, were those residues with peak broadening same in 3 cases? Did that support the conclusion that the site near T8 was involved in binding but not the C-terminal?

The peak broadening represented in Figure 2 was not obvious comparing to the signal noise. For the 2D spectrum, it would be more straightforward to represent an overlaid spectra of b and c to highlight the broadened peaks. 

It’s critical to represent the quantitative analysis of the NMR peak broadening, like the ratio of the peak width. Besides that, peak height is critical to suggest the intensity change was resulted from the peak broadening rather than concentration difference. Simply speaking I would recommend including two plots of peak width ratio over residues and peak intensity ratio over residues, which should be consistent with the conclusion of binding sites.

Author Response

Reply to reviewer #1
1.
Though O-glycan-dependent binding was observed in previous studies, I would
recommend comparing the binding of MUC1 with and without O
-glycan by NMR.
T
hank you for this suggestion. The experiment would further explore the role of O-glycan in
binding to
the antibody. Since we do not have unglycosylated peptide at this moment, we are
planning to do this experiment in future and the result will be included
in the next publication.
2.
If compared the spectra of the bound state of MUC1 in 3 different lengths, were those
residues with peak broadening same in 3 cases? Did that support the conclusion that the site

near T8 was involved in binding but not the C
-terminal?
W
e compared peak broadening in 3 different peptides and the results are shown in Figure 2c
and
Supplementary Figure S22 (main text line 146-150, 173-174). Overall, peak broadening
was observed for N
-terminal region of the peptides, suggesting that the region around T8 was
involved in binding.

3.
The peak broadening represented in Figure 2 was not obvious comparing to the signal noise.
For the 2D spectrum, it would be more straightforward to represent an overlaid spectra of b

and c to highlight the
broadened peaks.
We prepared a figure showing an overlaid spectra of b and c (Figure 2b)
, and the results are
described at line
146-150. Furthermore, Figure 3 is newly prepared showing NMR spectra of
MUC1 (20AA) highlighting the
V7 and T20 methyl signals with different MUC1 (20AA)-to-
antibody ratio.

4.
It’s critical to represent the quantitative analysis of the NMR peak broadening, like the ratio
of the peak width. Besides that, peak height is critical to suggest the intensity change was

resulted from
the peak broadening rather than concentration difference. Simply speaking I
would recommend including two plots of peak width ratio over residues and peak intensity

ratio over residues, which should be consistent with the conclusion of binding sites.

Than
k you very much for raising an important point. To analyze the data quantitatively, the
NMR peaks were evaluated in terms of peak height
only. This is because measuring NMR line
width is technically difficult
and resulted in significant errors. Therefore, we only show the
ratio of peak height
in Figure 2c and Supplementary Figure S22

Reviewer 2 Report

In this manuscript, the authors investigate the mode of the interactions between MUC1 O-glycopeptide and MY.1E12 using NMR ,molecular modeling and docking simulations. NMR titration results locates the epitope to the O-glycan and nearby amino acid residues. Furthermore, docking studies suggest that both O-glycan and peptide of the MUC1 bind to the MY.1E12 antibody.

Introduction section should be more developed. The methodology of the investigation of interaction between similar molecules should be also presented.

What kind of the interactions between two molecules are expected?

 Figure 1. The structure of the O-glycan should be provided .

 The quality of the 1H and COSY spectra (Fig. 2) are very poor. I assume this is due to the low concentrations and too short acquisition time ? How long the spectra were acquired? It is difficult to see significant differences between MUC1 signals without the presence of antygen and their mixture. Nevertheless, the most pronounced differences in signal intensity are seen for the H5 residue ( MUC1-27AA).

Please explain why you used such a low concentration? Perhaps the results of titration and COSY experiments for all MUC1 sequence would be better if the concentration of two molecules would be higher?

Have you considered STD NMR techniques in this study? Probably MUC1 (9AA) is too large molecule , but if you can reduce the amino acid chain of the MUC1 glycopeptide, it could be an excellent experiment to prove epitope localization. Is it possible?

Out of curiosity, please provide the molecular weight of the MUC1 glycopeptides and MY.1E12.1 antygen.

line 34:  transformation

Line93  Broader than those of

The materials and methods need to be improved. Details about frequencies of the spectrometers, probe type are not included. The detail of the performing the 1H NMR spectra ( number of scans, pulse length, temperature, presaturation ..) and 2D NMR spectra should be described.

Moreover, all full and partial ranges of the 1H NMR and 2D NMR spectra, titration experiments for all MUC1 sequences should be presented in supplementary materials. Please also provide the NMR assignment tables for the MUC1 glycopeptides

Author Response

Reply to reviewer #2
1
. Introduction section should be more developed. The methodology of the investigation of
interaction between similar molecules should be also presented.

We added
several examples analyzing interaction of similar molecules in the introduction
section (
line 43-51).
2
. What kind of the interactions between two molecules are expected?
To evaluate the mode of interaction between two molecules, a new figure (Figure 7d)
was
prepared to indicate the amino acid residues of antibody which are spatially close to the

glycopeptide antigen.
Since the model of antibody-antigen complex model may not be
accurate due to the lack of
full experimental information, we refrained from describing
detailed interaction modes.

3
. Figure 1. The structure of the O-glycan should be provided.
The
chemical structure of O-glycan is provided in Figure 1.
4
. The quality of the 1H and COSY spectra (Fig. 2) are very poor. I assume this is due to the
low concentrations and too short acquisition
time? How long the spectra were acquired? It is
difficult to see significant differences between MUC1 signals without the presence of
antigen
and their mixture. Nevertheless, the most pronounced differences in signal intensity are seen

for the H5 residue
(MUC1-27AA).
As
the reviewer pointed out, S/N was low. This is due to the limited amount of the ligands
and antibody
available. The COSY spectrum was acquired for 28 h.
5
. Please explain why you used such a low concentration? Perhaps the results of titration and
COSY experiments for all MUC1 sequence would be better if the concentration of two

molecules would be higher?

I agree
that higher concentration will provide better results. However, the amount of this
sample was limited, so we managed to perform NMR experiment using the limited sample.

6
. Have you considered STD NMR techniques in this study? Probably MUC1 (9AA) is too

large molecule, but if you can reduce the amino acid chain of the MUC1 glycopeptide, it could
be an excellent experiment to prove epitope localization. Is it possible?

We did not try STD
-NMR experiment because STD-NMR works best in D2O solution
(avoiding
unwanted spin diffusion from H2O signal). Another reason is that we could not find
a saturation point which is selective
against antibody. The chemical shift range of glycopeptide
is similar to
that of antibody. Due to these reasons, we did not try STD-NMR experiment.
Instead,
we are planning to perform water-LOGSY experiment which will work in H2O
solution, and the results will be described in future work.

7
. Out of curiosity, please provide the molecular weight of the MUC1 glycopeptides and
MY.1E12.1 antygen.

M
olecular weight of MUC1 glycopeptides is added in Figure 1. Molecular weight of MY.1E12
antibody
is added in the materials and method section (line 385).
8
. line 34: transformation, Line93: Broader than those of
The
se typos have been corrected, thank you.
9
. The materials and methods need to be improved. Details about frequencies of the
spectrometers, probe type are not included. The detail of the performing the 1H NMR spectra

(number
of scans, pulse length, temperature, presaturation.) and 2D NMR spectra should be
described.

W
e have included the details of NMR measurement conditions in the materials and methods
section
(line 371-372, 379-382).
10
. Moreover, all full and partial ranges of the 1H NMR and 2D NMR spectra, titration
experiments for all MUC1 sequences should be presented in supplementary materials. Please

also provide the NMR assignment tables for the MUC1 glycopeptides
.
1
H-NMR, 2D NMR, and titration experiment spectra for all regions were added as
Supplementary Figure
s. Assignment tables of the MUC1 glycopeptides were added as Supplementary Table S1-S3.

Round 2

Reviewer 2 Report

The manuscript after major revision is sufficiently improved and can be publish in IJMS.